# Epidermal Barrier Development via Corneoptosis: A Unique Form of Cell Death in Stratum Granulosum Cells

**DOI:** 10.3390/jdb11040043

**Published:** 2023-11-30

**Authors:** Takeshi Matsui

**Affiliations:** Laboratory for Evolutionary Cell Biology of the Skin, School of Bioscience and Biotechnology, Tokyo University of Technology, 1404-1, Katakura-cho, Tokyo 192-0982, Japan; matsuitks@stf.teu.ac.jp; Tel.: +81-42-637-2149

**Keywords:** epidermal development, stratum granulosum, stratum corneum, epidermal barrier, corneoptois, cornification

## Abstract

Epidermal development is responsible for the formation of the outermost layer of the skin, the epidermis. The establishment of the epidermal barrier is a critical aspect of mammalian development. Proper formation of the epidermis, which is composed of stratified squamous epithelial cells, is essential for the survival of terrestrial vertebrates because it acts as a crucial protective barrier against external threats such as pathogens, toxins, and physical trauma. In mammals, epidermal development begins from the embryonic surface ectoderm, which gives rise to the basal layer of the epidermis. This layer undergoes a series of complex processes that lead to the formation of subsequent layers, including the stratum intermedium, stratum spinosum, stratum granulosum, and stratum corneum. The stratum corneum, which is the topmost layer of the epidermis, is formed by corneoptosis, a specialized form of cell death. This process involves the transformation of epidermal keratinocytes in the granular layer into flattened dead cells, which constitute the protective barrier. In this review, we focus on the intricate mechanisms that drive the development and establishment of the mammalian epidermis to gain insight into the complex processes that govern this vital biological system.

## 1. Evolution from Amniotes to Mammals

The evolution from amniotes to mammals, which occurred over hundreds of millions of years, has entailed significant developments [1,2,3,4,5]. Amphibians gave rise to amniotes, a group that includes reptiles, birds, and mammals, approximately 360 million years ago (Figure 1). The defining feature of amniotes is the presence of an amniotic egg that provides extraembryonic protection to the developing embryo. Approximately 320 million years ago, a group of amniotes called synapsids appeared. Synapsids were characterized by the presence of a single temporal fenestra (hole) on each side of the skull behind the eye socket. Mammals eventually evolved from synapsids. A group of synapsids, known as therapsids, emerged approximately 275 million years ago. They had several mammal-like features, such as advanced jaw mechanics and internal thermoregulation. Some therapsids were large in size and could have been the dominant terrestrial animals of their time. Approximately 245 million years ago, cynodonts evolved from therapsids. They had mammal-like characteristics, such as a more upright posture, a larger brain, and hair-like structures on their skin, which could have been an early form of insulation. Approximately 225 million years ago, cynodonts gave rise to mammals [6]. Mammals are characterized by the presence of hair, mammary glands that produce milk for the young, and a four-chambered heart. They also have specialized teeth to process food more efficiently. Since their emergence, mammals have diversified into various forms, from tiny shrews to massive whales, and have become dominant land animals occupying almost every ecological niche.

Epidermal development was observed during the amphibian stage of evolution (Figure 1). The evolution of the epidermis from amphibians to mammals is a complex process that has occurred over millions of years. Although the basic structure of the epidermis remains similar across vertebrates, its function and complexity differ significantly among species. As vertebrates continued to evolve, the epidermis became more specialized with the development of structures such as hair, feathers, and scales. In mammals, the epidermis has undergone further modifications, including the development of sweat and sebaceous glands, which help regulate body temperature and maintain skin hydration.

## 2. Epidermal Development: Formation of Stratified Squamous Epithelia from Simple Epithelia

During mammalian embryogenesis, the skin epidermis originates from the ectoderm [7] (Figure 2). First, the ectoderm, a monolayer of epithelial tissue expressing keratins 8 and 18, forms on the stratum basale (basal layer). Analysis of p63 knockout mice has revealed that p63 drives a signaling pathway essential for epidermal formation [8,9]. Moreover, during this period, an additional layer called the periderm is formed above the basal layer. This layer constitutes a transient barrier that is shed during the subsequent process of multilayered squamous epithelialization [10]. In support of this, the tight junctional protein claudin-6 is expressed in the uppermost layer of the periderm [11]. During embryogenesis, basal cells in the stratum basale undergo asymmetric cell division to form an intermediate layer, the spinous layer (stratum spinosum), which rapidly forms a multilayered structure [12]. The stratum spinosum further differentiates to form the stratum granulosum (SG). The stratum corneum (SC), a layer of dead cells, is formed after the shedding of the periderm, which differentiates specifically from the granular layer via cell death of SG cells. Epidermal formation occurs during exposure to amniotic fluid, indicating that the epidermal barrier function is complete at birth before exposure to the external environment [13]. Through these organized architectural construction steps, the epidermis functions as a crucial barrier in the mammalian body, enabling survival in a terrestrial environment.

## 3. Evolution of the Mammalian Stratum Corneum

Several skin-specific proteins are differentially expressed during stratification of the mammalian epidermis, most of which are located in the epidermal differentiation complex (EDC) on chromosome 1q21 in humans and chromosome 3 in mice [14,15]. The EDC has undergone significant evolution in the mammalian lineage [16]. Skin aspartic protease (SASPase/Taps/ASPRV1) is a retrovirus-like, skin-specific aspartic protease that is specifically expressed in the uppermost SG [17,18,19] and is similar to equine anemia virus protease. Mutations causing decreased autoprocessing activity of human SASPase were found in a Japanese cohort of patients with atopic dermatitis (AD), although these mutations were not associated with AD or clinically dry skin in European populations [20,21]. These SASPase mutations have been reported in ichthyosis vulgaris in humans and dogs [22,23]. Human SASPase cleaves the linker sequences of profilaggrin to produce monomeric filaggrin in vitro [20]. SASPase knockout mice exhibit accumulation of aberrantly processed profilaggrin in the lower SC, an increase in the number of SC layers, and decreased SC hydration, that is, dry skin, but no significant change in trans-epidermal water loss or natural moisturizing factor [20]. SASPase and filaggrin emerged with the evolution of mammals. It has been suggested that the SASPase gene was integrated into the ancestral mammalian genome either via retroviral infection or transposition of retroviral elements, which serves as a typical example of exaptation (co-option). SASPase subsequently began to cooperate in regulating the moisturization of the SC [24] (Figure 1). The mechanism by which the aspartic protease SASPase activates and cleaves filaggrin in the lower layers of the SC was not well understood until recently. In this review, we discuss the potential mechanisms that activate SASPases.

## 4. Cornification: The Final Stage of Epidermal Differentiation

In the epidermis of adult mammalian skin, keratinocytes arise from the proliferative basal layer, differentiate, and migrate upward to form several spinous layers [25] (Figure 3). They form three granular layers (SG) (called SG1-3 from the superficial layer) with a flattened Kelvin’s tetrakaidecahedron cell shape in mice and humans [26,27]. Tight junctions (TJs) are formed in the SG2 cell layer [28,29,30]. SG1 cells above the TJ undergo cell death, and a functional air–liquid interface barrier, that is, the SC, is formed. Intracellular morphological changes in dying SG1 cells allow for the formation of a functional barrier. Although “cornification” refers to the entire process of SC formation, the first step in cornification is corneoptosis of SG1 cells, a unique mode of cell death [31,32,33]. Corneoptosis is followed by five major phenomena: keratohyalin granule disappearance, organelle degradation, formation of a dense keratin network, formation of lamellar lipids, and formation of the cornified envelope (Figure 3 and Figure 4).

Keratohyalin granules, which are composed of profilaggrin protein via liquid–liquid phase separation, disappear just after the cell death of SG1 cells [34,35]. Degradation of all organelles, including the nuclear membrane, nuclear DNA, mitochondria, Golgi apparatus, endoplasmic reticulum, and lysosomes, occurred after SG1 cell death. In particular, just before cell death, mitochondria are fragmented [36], and the degradation of mitochondria in SG1 is regulated by the mitophagy receptor, NIX, as demonstrated in live imaging of a human epidermal equivalent model [37,38]. In pathological conditions such as AD and psoriasis, the nuclei remain in the SC, suggesting an alteration in this degradation mechanism in these diseases. Keratin filaments are abundantly expressed in the epidermal keratinocytes. Keratins 5 and 14 are expressed in the stratum basale and form heterodimers, whereas keratins 1 and 10 are expressed in the differentiated layer of the epidermis. These epidermal keratins are bundled into the cytoplasm and contribute to the formation of a dense keratin network in the SC. Lamellar lipids are composed of ceramide, cholesterol, and fatty acids. These are biosynthesized from precursor lipids, glucosylceramide, cholesterol, and phospholipids by lipid-metabolizing enzymes, such as β-glucocerebrosidase and sphingomyelinase, secretory phospholipase A_2_ during the secretion of specialized secretory vesicles, and lamellar granules which are gradually formed from the suprabasal keratinocytes (see details in [39,40,41,42]). The uniqueness of lamellar lipids lies in their molecular architecture. Cryo-electron microscopic analysis revealed that cholesterol and fatty acids are associated with each side chain of the extended ceramide structure [43]. The plasma membrane of SG1 cells transforms into a cornified envelope, which contains highly cross-linked insoluble proteins below the plasma membrane via transglutaminase (TGase) activity [44]. TGases are enzymes activated by Ca^2+^ that catalyze the formation of covalent bonds between the γ-carboxamide groups of glutamine residues and ε-amino groups of lysine residues in primary amines, peptides, and proteins [4,45]. Through these five unique processes, corneocytes and dead cells constitute the SC. Subsequently, corneocytes are modified by further biochemical reactions and mature to act as functional barriers.

## 5. Corneoptosis: A Unique Type of Cell Death in SG1 Cells

Cell death in SG1 is a complex and unique phenomenon. Recently, through intravital imaging of mouse SG1 cells, we demonstrated that SG1 cell death comprises two phases: phase I and II (Figure 5) [32]. (i) In phase I there is an increase in [Ca^2+^]_i_ in SG1 cells under neutral pH conditions for approximately 60 min, followed by (ii) phase II in which there is a rapid decrease in intracellular pH to a weakly acidic state while [Ca^2+^]_i_ remain high. (iii) Under phase II conditions, keratohyalin granules, nuclear DNA, and all organelles are degraded and transformed into corneocytes. Subsequently, they migrate upward, eventually reaching the surface, and are shed off (Figure 5).

Cell death is a physiological phenomenon that is essential for the development and homeostasis of tissues and organs [46]. To date, many studies have investigated endogenous/exogenous apoptosis, mitochondrial permeability transition, driven necrosis, necrosis, ferroptosis, pyroptosis, parternatosis, entosis cell death, NETotic cell death (neutrophil extracellular traps (NET)), lysosome-dependent cell death, autophagy-dependent cell death, immunogenic cell death, and various other cell deaths, which have been classified based on their molecular and morphological characteristics [47]. During these types of cell death, the remnants of dead cells are eliminated by macrophages and adjacent cells through phagocytosis via efferocytosis. If the removal of dead cells is uncontrolled, cellular content, mitochondria, and DNA leakage from dead cells can cause tissue inflammation. Thus, cell death terminates the original physiological function of the cell, and the dead cell body is unnecessary for the organism. However, a characteristic feature of SG1 cell death is that dead cells (corneocytes) are not eliminated but rather function as essential components of the SC barrier. The SC allows terrestrial vertebrates to survive in gaseous environments by protecting the body from mechanical stress, pathogenic invasion, toxic substances, and dehydration [4]. We termed this concept of functional cell death of SG1 cells that occurs in the early stages of cornification as “corneoptosis” [32,33].

## 6. Physiological Role of Phase I of Corneoptosis: Long-Lasting Intracellular Ca^2+^ Elevation in SG1 Cells

Various types of [Ca^2+^]_i_ signals have been reported. The duration of [Ca^2+^]_i_ elevation varies from milliseconds to minutes in different cell types and may occasionally oscillate. In vivo, [Ca^2+^]_i_ reported in neurons and other somatic cells is often short, spike-like [Ca^2+^]_i_ spurts lasting milliseconds. In the case of SG1 cell death, [Ca^2+^]_i_ elevation persisted throughout the SC even after cell death (Figure 4). This raises the question of the physiological role of the prolonged (approximately 60 min) increase in [Ca^2+^]_i_ observed in phase I corneoptosis, which occurs under neutral-pH conditions. In general, the rapid rise in excessive [Ca^2+^]_i_, referred to as [Ca^2+^]_i_ overload, is a phenomenon also observed in other forms of cell death, such as apoptosis, necrosis, autophagic cell death, pyroptosis, and NETosis [48,49,50,51,52]. Prolonged elevation of [Ca^2+^]_i_ is a unique feature of SG1 cells, which occurs just once and persists even after they transform into corneocytes after death. It is speculated that long-term increases in [Ca^2+^]_i_ levels in SG1 cells induce activation or conformational changes in various Ca^2+^-dependent enzymes such as caspases, calpains, and other proteases, resulting in the elevation of their activity. TGase is a major Ca^2+^-activating enzyme that catalyzes the covalent cross-linking of proteins through the formation of isopeptide bonds, which require a neutral pH for optimal activity [45]. The cornified envelope (CE) is an insoluble structure of keratinocytes formed from the plasma membranes. Elevated [Ca^2+^]_i_ in SG1 cells may activate TGase-1 and TGase-3, allowing them to cross-link keratinocyte-specific proteins such as keratin 1/10, involucrin, periplakin, envoplakin, and loricrin, thereby forming a rigid CE underneath the plasma membrane [44]. Thus, maintaining high [Ca^2+^]_i_ at neutral pH for extended periods, as seen in phase I in SG1 cells, may be important for achieving a complete cross-linking reaction at the site of CE formation. However, it is unclear whether 60 min of phase I corneoptosis is sufficient to form a mature CE.

## 7. Physiological Role of Phase II of Corneoptosis: Intracellular Acidification of SG1 Cells

After persistent and prolonged elevation in [Ca^2+^]_i_ levels in mouse SG1 cells for approximately 60 min, the intracellular pH (pH_i_) immediately dropped from neutral to weakly acidic (Figure 4). This dramatic change in pH_i_ did not affect elevated [Ca^2+^]_i_ levels. Thus, [Ca^2+^]_i_ level is thought to remain high throughout the SC. The exact mechanism responsible for this immediate drop in pH_i_ remains unclear. We isolated mouse SG1 cells and showed that plasma membrane disruption, i.e., plasma membrane permeability, increases after [Ca^2+^]_i_ elevation. Thus, it is possible that permeabilization of the plasma membrane may lead to the incorporation of extracellular protons into SG1 cells. If this is the case, the extracellular pH of SG1 cells must be acidic, although the exact pH value of the extracellular pH of SG1 cells above TJs has not been clarified. Under these unusual intracellular conditions (high [Ca^2+^]_i_ and acidic pH_i_), corneocyte-specific intracellular changes can occur.

KHGs consist of aggregated protein structures, such as profilaggrin (a precursor of filaggrin), loricrin, and trichohyaline. Quiroz et al. demonstrated that a decrease in intracellular pH < 6.2 in SG1 cells induces the loss of KHGs in vivo using intravital imaging of neonatal mice [35]. We also showed that the disappearance of KHGs from isolated mouse SG1 cells is highly sensitive to the pH of the medium [32]. These in vivo and in vitro data suggest that KHGs disappear and become corneocytes in a manner that is directly coordinated with intracellular acidification in SG1 cells during phase II.

During or after the disappearance of KHGs, the cleavage of profilaggrin to filaggrin may be another important event in the transition from dying SG1 cells to corneocytes. Filaggrin is a barrier-related protein that is specifically expressed in the granular layer of the mammalian epidermis [53]. A loss-of-function mutation in the human filaggrin gene is also a major predisposing factor for AD in humans [54]. Moreover, filaggrin-deficient mice exhibit aberrant SC barrier functions [55]. Profilaggrin, which consists of an N-terminal, two Ca^2+^ binding domains, a tandemly linked filaggrin monomer (10–12 in humans), and a C-terminal domain, forms KHGs. As outlined in the previous paragraph, profilaggrin is cleaved at its linker sequence by specific proteases. One such enzyme is SASPase, a retrovirus-like aspartic protease that acts during the transition of SG1 cells to corneocytes [20]. Because the optimum pH for protease activity of this enzyme is slightly acidic (pH 5.77 for mouse and pH 6.27 for human SASPase), cleavage of the linker sequence of profilaggrin may occur during acidification [19,56]. The production of monomeric filaggrin in lower SC cells may contribute to the bundling of keratin filaments [57,58].

In addition to the disappearance of KHGs within the SC, all organelles underwent degradation. This is also thought to occur under corneoptosis phase II conditions. As lysosomes are also absent from the SC, proteases contained in the lysosome are thought to be released into the cytoplasm during lysosome rupture during corneoptosis. Most organelles are thought to be degraded by lysosomal enzymes activated under acidic conditions. In addition, a large amount of nuclear DNA is packed inside the nucleus, which is the largest organelle in the SG1 cells. The DNA degradation rate is susceptible to the pH of the extracellular medium, suggesting the involvement of DNases that are activated under acidic conditions. DNA degradation in isolated SG1 cells is affected by the pH of the culture medium [32]. DNase activation under weakly acidic conditions may be involved in the degradation of large amounts of nuclear DNA. This concept is supported by the fact that the major DNases expressed in differentiated epidermal layers, such as DNase1-like 2 (DNase1L2) and DNase2 [59,60], require an acidic pH for optimum activity [61,62]. Furthermore, the degradation of nuclear DNA during cornification has been shown to be defective in mouse skin deficient in DNase1L2 and DNase2 genes [63].

Through these intracellular morphological changes, the cytoplasmic space of SG1 cells becomes free of KHGs and organelles during phase II corneoptosis. Corneocytes must be flattened to a depth of 2 µm from that of 5 µm of the SG1 cells, resulting in the filling of a dense keratin network into this space to provide corneocytes with mechanical strength.

## 8. Future Prospects for Corneoptosis Research

It is anticipated that elucidating the corneoptosis process will yield a more thorough understanding of SC formation. Nevertheless, certain aspects of corneoptosis remain unresolved. For example, it is unclear whether it also occurs in the epidermis of all terrestrial vertebrates, especially during embryonic development, since studies have only examined it in adult mouse SG1 cells. SG1 cells in normal skin are believed to undergo neither apoptosis nor necrosis and express genes that inhibit pyroptosis. The molecular mechanisms by which SG1 cells shift from corneoptosis to other forms of cell death under pathological conditions or stress remain unclear. Barrier dysfunction in TJ and SC has been reported in AD [30]. In AD, the pH of SC is reported to be elevated between 0.1–0.9 units [64]. Congenital diseases such as keratitis-ichthyosis-deafness (KID) syndrome and Darier’s disease display dysregulated Ca^2+^ signaling, leading to aberrant SC formation [65,66]. Under these pathological conditions, the occurrence of corneoptosis phases I and II may be affected spatially and temporally. It is thought that the TJs formed on SG2 cells have both double- and single-TJ strand formation according to the turnover stage of SG2 cells, but the nature of its association with the induction of corneoptosis is uncertain. Additionally, it is uncertain whether the cellular biomedical properties of SG1 cells induce corneoptosis, which is conserved across terrestrial vertebrates. Therefore, a detailed cell biology study of corneoptosis would elucidate the mechanisms of environmental adaptation in mammalian skin with respect to cell death.

## Figures and Tables

**Figure 1 jdb-11-00043-f001:**
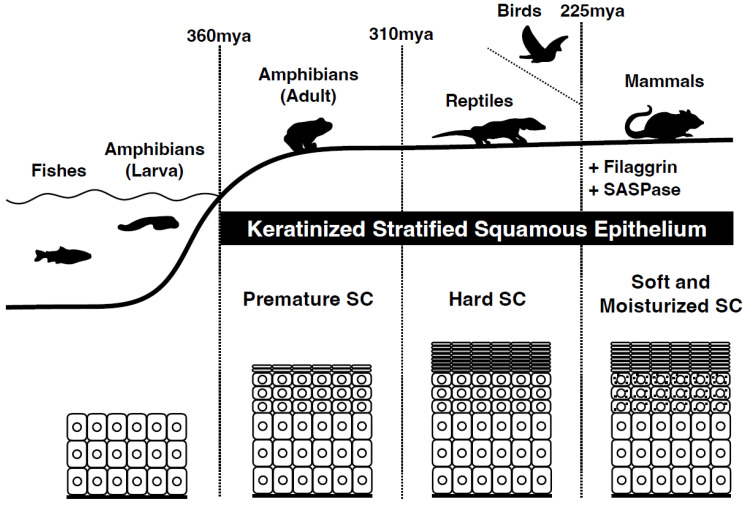
Evolution from amniotes to mammals. The emergence of the first terrestrial vertebrate amphibians (e.g., ichthyostega) from water and their adaptation to life on land occurred approximately 360 million years ago (mya) during the late Devonian period. They underwent an evolutionary change in their surface epithelium, from multilayered epithelia to keratinized stratified squamous epithelia as a protective measure against water loss and sunlight. Reptiles are thought to have developed a stiff stratum corneum 310 million years ago. The first mammals appeared 225 million years ago. These mammals acquired a soft and moist stratum corneum. Genome analysis of different species of terrestrial vertebrates has shown the integration of new genes specific to stratified epithelia, such as filaggrin and SASPase. The origin of SASPase can be traced back to an ancient retroviral infection as an example of exaptation.

**Figure 2 jdb-11-00043-f002:**
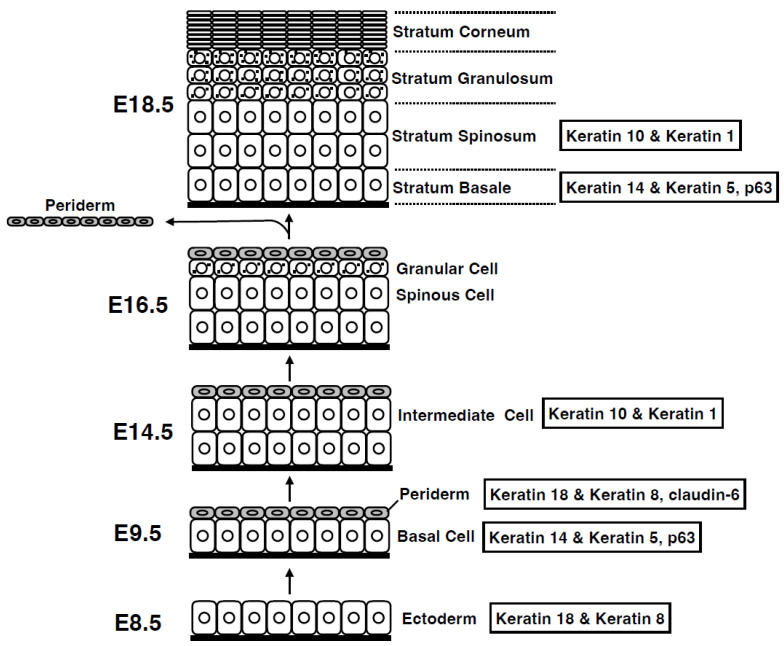
Development of the epidermis. During development, the epidermis forms via vertical stratification. Initially, a single layer of ectoderm-derived cells on the basement membrane differentiates into basal layer cells. Basal layer cells produce both the periderm on the top and the intermediate layer between the periderm and the basal layer. The intermediate cells then differentiate into spinous cells, which further differentiate into granular cells. The periderm is shed during the later stages of embryogenesis, and the granular cells differentiate into cornified cells before birth. Embryonic days (E) are based on mouse embryogenesis.

**Figure 3 jdb-11-00043-f003:**
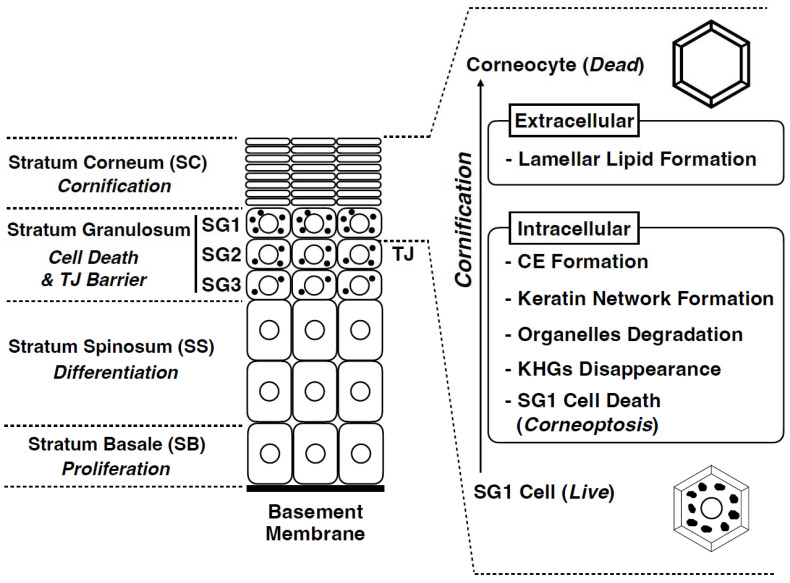
Structure of the mammalian epidermis and major cornification processes. The epidermis is composed of the following four cell layers: stratum basale (SB), stratum spinosum (SS), stratum granulosum (SG), and stratum corneum (SC). The SG is composed of three cell layers (SG1, SG2, and SG3 from the outside to the inside). Tight junctions (TJs) are formed in the SG2 cells. Cornification comprises several unique intracellular and extracellular processes. Corneoptosis is a unique type of cell death of stratum granulosum cells (SG1 cells). It is followed by keratohyalin granules disappearance, organelle degradation, dense keratin network formation, cornified envelope formation, and lamellar lipid formation.

**Figure 4 jdb-11-00043-f004:**
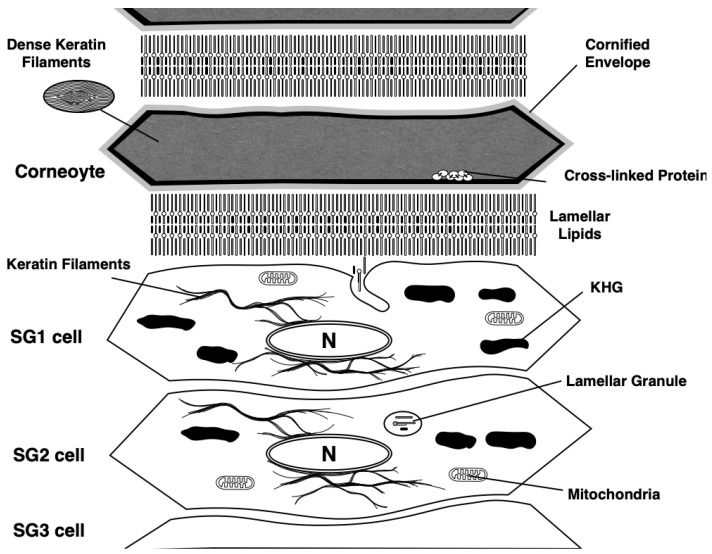
Coneoptosis occurs at the initial stage of cornification. After the cell death, keratohyalin granules disappearance and organelle degradation are induced. Dense keratin networks are formed in the cytoplasm of dead corneocytes. Plasma membranes are changed into cornified envelopes, which are formed by crosslinking of various hydrophobic proteins, such as periplakin, epiplakin, loricrin, small proline-rich protein family, and filaggrin via the activity of transglutaminases. Lamellar lipids are formed at the extracellular space of corneocytes, which are composed of ceramide, fatty acid, and cholesterol. KHG: Keratohyalin granule.

**Figure 5 jdb-11-00043-f005:**
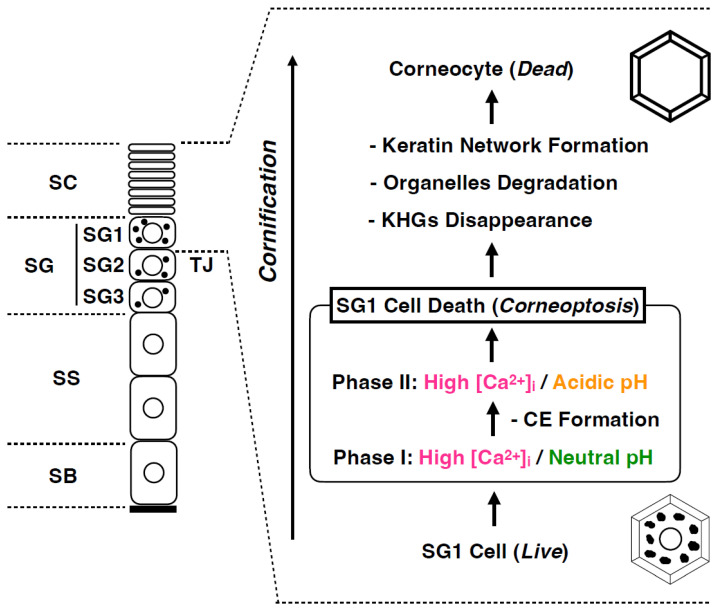
Functional cell death: Corneoptosis. SG1 cell death involves a sequence of distinct intracellular ionic alterations: Phase I is characterized by elevated Ca^2+^ levels and a neutral pH sustained for roughly 60 min; and Phase II is marked by persistently high Ca^2+^ concentration coinciding with a shift to an acidic pH. These unique ionic modifications precipitate the dissolution of keratohyalin granules, the subsequent degradation of nuclear DNA, and the systematic breakdown of organelles, culminating in the transformation of viable SG1 cells into corneocytes that perform essential barrier functions.

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
