# Peer review of "Epidermal Barrier Development via Corneoptosis: A Unique Form of Cell Death in Stratum Granulosum Cells"

_jdb, 2023, doi:10.3390/jdb11040043_

Round 1
Reviewer 1 Report
Comments and Suggestions for Authors
This article reviews (Manuscript ID: jdb-2507609) the establishment of the mammalian epidermis. It first introduces the evolutionary role of mammal skin in epithelial formation, with different molecule expressions. Differentially expressed genes contribute to epidermal stratification and cornification. The article then focuses on SG1 functional cell death, which is termed corneoptosis, and demonstrates various Ca2+ oscillations.
Overall, the topic is interesting, and the writing is well-structured. However, there are several instances where the figures do not align with the corresponding text.
Major comments:
l There are inconsistencies between the figures and the text in several sections of the article.
o Figure 1 should be associated with the paragraph titled "Evolution from amniotes to mammals.”
o Line 52: Figure 1 should be changed to Figure 2 to match the content.
o Line 107: Figure 2 should be changed to Figure 3, which illustrates the three layers of the stratum granulosum in the mammalian epidermis.
o Line 118: Figure 2 should be changed to Figure 4, which illustrates the 5 phenomena of corneoptosis.
o It seems that lines 155, 160, 165, 175, and 178 of "Figure 3" do not correspond with the text.
l To better illustrate the details of the two phases of corneoptosis, it would be nice to use a different schematic diagram.
Minor comments:
l In the figure "Development of the epidermis," it would be better to label different molecules as keratin 8, 18, p63, and claudin-6.
l In line 116-118, five phenomena of corneoptosis were described and each term was explained in the following paragraph. However, the "keratohyalin granules disappearance" was missing.
l Misspelling of ‘Lamellar’ lipid formation in figure 4.
Author Response
I am grateful for the encouraging comments on my manuscript. I deeply appreciate the insightful and constructive comments provided by the reviewers. I believe your inputs have helped us improve the quality of my review article.Major comments: There are inconsistencies between the figures and the text in several sections of the article.
Thank you very much for pointing out the inconsistency between the figures and the text.
- Figure 1 should be associated with the paragraph titled "Evolution from amniotes to mammals.
I changed the title of Figure 1 to "Evolution from amniotes to mammals.” (Line 48, page 2).
- Line 52: Figure 1 should be changed to Figure 2 to match the content.
I corrected “Figure 1” and “Figure 2” in several sentences (Line 30, page1; Lines 59 and 70, page 2).
- Line 107: Figure 2 should be changed to Figure 3, which illustrates the three layers of the stratum granulosum in the mammalian epidermis.
I corrected “Figure 2” to “Figure 3” (Lines 121 and Line 131, page 3).
- Line 118: Figure 2 should be changed to Figure 4, which illustrates the 5 phenomena of corneoptosis.
I corrected “Figure 2” to “Figure 4” (Lines 177 and 182, page 4; Lines 215 and 240, page 5).
- It seems that lines 155, 160, 165, 175, and 178 of "Figure 3" do not correspond with the text.
Thank you very much for pointing out the incorrect numbering of the figures. We have corrected the figure numbers as you described above.
- To better illustrate the details of the two phases of corneoptosis, it would be nice to use a different schematic diagram.
I have added Figure 5, entitled “Functional cell death, corneoptosis” to illustrate the details of the two phases of keratinocyte death (Lines 184-190, page 4).
- In the figure "Development of the epidermis," it would be better to label different molecules as keratin 8, 18, p63, and claudin-6.
I have labeled keratin 8, 18, p63, and claudin-6 in the cartoon of figure 2.
- In line 116-118, five phenomena of corneoptosis were described and each term was explained in the following paragraph. However, the "keratohyalin granules disappearance" was missing.
I have added sentence “Keratohyalin granules, which are composed of profilaggrin protein via liquid-liquid phase separation, disappear just after the cell death of SG1 cells [34,35].” (Lines 148-149, page 4).
- Misspelling of ‘Lamellar’ lipid formation in figure 4.
I have corrected the spelling of ‘Lamellar’ in new Figure 3.
Reviewer 2 Report
Comments and Suggestions for Authors This manuscript reviews the process of skin epidermal barrier development with a specific focus on corneoptosis, a specialized form of cell death. The topic is very interesting, and the review is well-written. It will be more helpful if the authors can discuss the potential clinical impact of skin epidermal barrier development via corneoptosis. Moreover, a summarized diagram (figure) showing the physiological roles of phase I and II of corneoptosis would be helpful for readers to understand the process. Minor points: There is a couple of typo error. For instance, LINE 59: this layer constitutes a transient a barrier.Author Response
I am grateful for the encouraging comments on my manuscript. I deeply appreciate the insightful and constructive comments provided by the reviewers. I believe your inputs have helped us improve the quality of my review article.
This manuscript reviews the process of skin epidermal barrier development with a specific focus on corneoptosis, a specialized form of cell death. The topic is very interesting, and the review is well-written.
I would like to thank the reviewer for your valuable comments.
- It will be more helpful if the authors can discuss the potential clinical impact of skin epidermal barrier development via corneoptosis.
Thank you for your insightful comment. In response, I have added sentences to address the the clinical impact of corneoptosis studies. These are “Barrier dysfunction in TJ and SC has been reported in AD [30]. In AD, the pH of SC is reported to be elevated between 0.1–0.9 units [63]. Congenital diseases such as keratitis-ichthyosis-deafness (KID) syndrome and Darier’s disease display dysregulated Ca2+ signaling, leading to aberrant SC formation [64,65]. Under these pathological conditions, the occurrence of corneoptosis phases I and II may be affected spatially and temporally.” (Lines 305-310, page 7).
- Moreover, a summarized diagram (figure) showing the physiological roles of phase I and II of corneoptosis would be helpful for readers to understand the process.
Thank you for your helpful suggestion. I have added Figure 5, entitled “Functional cell death, corneoptosis” to illustrate the details of the two phases of keratinocyte death (Lines 184-190, page 4).
- Minor points: There is a couple of typo error. For instance, LINE 59: this layer constitutes a transient a barrier.
I have corrected the phrase ‘a transient a barrier’ to ‘a transient barrier’ (Line 74, page 2). Additionally, I corrected several typographical errors throughout the manuscript, thanks to the English editing service (highlighted by yellow).
Reviewer 3 Report
Comments and Suggestions for Authors
The author provides a very good overview of a complex and emerging field. The article is clearly written and the data is well chosen and supports the conclusions. The research is important from the basic science and biomedical perspective and recent research has provided new insights into mechanisms of skin development. Thus, the review is relevant and timely.
I have few comments:
1) Generally, the manuscript does not include a lot of references. Even statements referring to specific research do not always show sufficient citations. Moreover, several references are rather old; obviously these can be important and valuable, but further publications from a fast changing area should be used.
2) Figures are not completely convincing. In part, they are redundant. On the other hand, especially figures 3 and 4, which also refer to author's own work, do not contribute to the details that are described in the text with references to the figures. The concept of illustrations for the article should be revised.
3) The text gives a brief overview of morphological characteristics of the integument in rather distant species. It does not clearly point to differences between mammalia. In particular, skin structure in rodents and humans is obviously different; it could be more clearly identified if work has been done in mouse skin/mouse keratinocytes. Any differences to findings from humans, possibly including more recent publications as mentioned above, should be pointed out and discussed if possible.
Author Response
I am grateful for the encouraging comments on my manuscript. I deeply appreciate the insightful and constructive comments provided by the reviewers. I believe your inputs have helped us improve the quality of my review article.
The author provides a very good overview of a complex and emerging field. The article is clearly written and the data is well chosen and supports the conclusions. The research is important from the basic science and biomedical perspective and recent research has provided new insights into mechanisms of skin development. Thus, the review is relevant and timely.
Thank you for your helpful comments.
- Generally, the manuscript does not include a lot of references. Even statements referring to specific research do not always show sufficient citations. Moreover, several references are rather old; obviously these can be important and valuable, but further publications from a fast changing area should be used.
Thank you for your valuable comments. I have added more recent references related to the epidermal barrier formation as follows:
6. Cabreira, S.F.; Schultz, C.L.; da Silva, L.R.; Lora, L.H.P.; Pakulski, C.; do Rego, R.C.B.; Soares, M.B.; Smith, M.M.; Richter, M. Diphyodont tooth replacement of Brasilodon-A Late Triassic eucynodont that challenges the time of origin of mammals. J Anat 2022, 241, 1424-1440, doi:10.1111/joa.13756.
16. Strasser, B.; Mlitz, V.; Hermann, M.; Rice, R.H.; Eigenheer, R.A.; Alibardi, L.; Tschachler, E.; Eckhart, L. Evolutionary origin and diversification of epidermal barrier proteins in amniotes. Mol Biol Evol 2014, 31, 3194-3205, doi:10.1093/molbev/msu251.
27. Yoshida, K.; Yokouchi, M.; Nagao, K.; Ishii, K.; Amagai, M.; Kubo, A. Functional tight junction barrier localizes in the second layer of the stratum granulosum of human epidermis. J Dermatol Sci 2013, 71, 89-99, doi:10.1016/j.jdermsci.2013.04.021.
34. Avecilla, A.R.C.; Quiroz, F.G. Cracking the Skin Barrier: Liquid-Liquid Phase Separation Shines under the Skin. JID Innov 2021, 1, 100036, doi:10.1016/j.xjidi.2021.100036.
36. Ipponjima, S.; Umino, Y.; Nagayama, M.; Denda, M. Live imaging of alterations in cellular morphology and organelles during cornification using an epidermal equivalent model. Sci Rep 2020, 10, 5515, doi:10.1038/s41598-020-62240-3.
37. Simpson, C.L.; Tokito, M.K.; Uppala, R.; Sarkar, M.K.; Gudjonsson, J.E.; Holzbaur, E.L.F. NIX initiates mitochondrial fragmentation via DRP1 to drive epidermal differentiation. Cell Rep 2021, 34, 108689, doi:10.1016/j.celrep.2021.108689.
38. Zaver, S.A.; Johnson, C.J.; Berndt, A.; Simpson, C.L. Live Imaging with Genetically Encoded Physiologic Sensors and Optogenetic Tools. J Invest Dermatol 2023, 143, 353-361 e354, doi:10.1016/j.jid.2022.12.002.
40. Yamanishi, H.; Soma, T.; Kishimoto, J.; Hibino, T.; Ishida-Yamamoto, A. Marked Changes in Lamellar Granule and Trans-Golgi Network Structure Occur during Epidermal Keratinocyte Differentiation. J Invest Dermatol 2019, 139, 352-359, doi:10.1016/j.jid.2018.07.043.
41. Ishida-Yamamoto, A.; Yamanishi, H.; Igawa, S.; Kishibe, M.; Kusumi, S.; Watanabe, T.; Koga, D. Secretion Bias of Lamellar Granules Revealed by Three-Dimensional Electron Microscopy. J Invest Dermatol 2023, 143, 1310-1312 e1313, doi:10.1016/j.jid.2023.03.1674.
I have also added an important reference related to the definition of ethe volutionary term, “exaptation” as follows:
24. Gould, S.J., Vrba, E.S. Exaptation; a missing term in the science of form. Paleobiology 1982, 8, 4–15.
- Figures are not completely convincing. In part, they are redundant. On the other hand, especially figures 3 and 4, which also refer to author's own work, do not contribute to the details that are described in the text with references to the figures. The concept of illustrations for the article should be revised.
Thank you for your valuable suggestions. I have changed Figure 3 and 4 to better illustrate the concept of cornification and corneoptosis as follows:
“Figure 3. Structure of the mammalian epidermis and major cornification processes.” (Lines 132-139, page 3).
“Figure 4. Functional cell death, corneoptosis.” (Lines 140-146, page 3).
- The text gives a brief overview of morphological characteristics of the integument in rather distant species. It does not clearly point to differences between mammalia. In particular, skin structure in rodents and humans is obviously different; it could be more clearly identified if work has been done in mouse skin/mouse keratinocytes. Any differences to findings from humans, possibly including more recent publications as mentioned above, should be pointed out and discussed if possible.
Thank you for your helpful comments. I have specified the species names used for each set of experimental data (Lines 103, 107, 120 and 123, page3; Line 238, page 5; Line 254 and 262, page 6). Additionally, I have added the following sentences; Nevertheless, certain aspects of corneoptosis remain unresolved. For example, it is unclear whether it also occurs in the epidermis of all terrestrial vertebrates, especially during embryonic development, since studies have only examined it in adult mouse SG1 cells.” (Lines 299-302, page 6).
Reviewer 4 Report
Comments and Suggestions for Authors
This is a very in-deapth treview on the subject i.e. epidermal barrier development via corneoptosis, to me deserves publication in a present form.
Author Response
Comment from Reviewer #4:
This is a very in-deapth review on the subject i.e. epidermal barrier development via corneoptosis, to me deserves publication in a present form.
Thank you very much for your encouraging comments.
Round 2
Reviewer 3 Report
Comments and Suggestions for Authors
The authors have thoroughly revised the manuscript and answered most questions. I have few more comments:
The separate illustrations of morphological and functional aspects of terminal differentiation (figures 3-5) are helpful. I would suggest to make some changes to figure 4, showing parts of morphological changes:
- The figure implies that short pieces of keratin filaments are available in the SG1 cell and short dense filaments in the SC.
- Moreover, the distribution of components seems to be quite arbitrary. However, most of these compounds are highly organised and namely filaments are part of the essential extended structures of the cytoskeleton.
- Suprabasal keratinocytes would have to produce and provide components of the cornified and lipid envelopes; though extrusion of lipids is indicated, this should be clarified.
- Figure 5 is not referred to in the main text; references in section 5 should probably refer to figure 5 (instead of figure 4).
Author Response
Responses to Reviewer’s Comment
Comment from Reviewer #3:
The authors have thoroughly revised the manuscript and answered most questions. I have few more comments
The separate illustrations of morphological and functional aspects of terminal differentiation (figures 3-5) are helpful. I would suggest to make some changes to figure 4, showing parts of morphological changes:
Thank you very much for your helpful comments on the figure 4 and the text.
- The figure implies that short pieces of keratin filaments are available in the SG1 cell and short dense filaments in the SC. Moreover, the distribution of components seems to be quite arbitrary. However, most of these compounds are highly organized and namely filaments are part of the essential extended structures of the cytoskeleton.
I modified the artwork in Figure 4. The pattern s of keratin filaments in the SG1, SG2 and the corneocyte were altered. I drew long bundled keratin filaments in SG1/SG2 cells and dense keratin filaments in the corneocyte.
- Suprabasal keratinocytes would have to produce and provide components of the cornified and lipid envelopes; though extrusion of lipids is indicated, this should be clarified.
I included SG2 cells alongside SG1 cells to demonstrate the precursors of lamellar lipids, which are prepared in the lamellar body. Additionally, I added sentences to explain the preparation, extrusion, and formation of lamellar lipids, citing additional reference [42].
I added following sentences; “Lamellar lipids are composed of ceramide, cholesterol, and fatty acids. These are biosynthesized from precursor lipids, glucosylceramide, cholesterol and phospholipids by lipid-metabolizing enzymes, such as β-glucocerebrosidase and sphingomyelinase, secretory phospholipase A2 during the secretion of specialized secretory vesicles, lamellar granules which are gradually formed from the suprabasal keratinocytes (see details in [39-42])” (Lines 160 - 164, page 4).
[42] Norlen, L.; Lundborg, M.; Wennberg, C.; Narangifard, A.; Daneholt, B. The Skin's Barrier: A Cryo-EM Based Overview of its Architecture and Stepwise Formation. J Invest Dermatol 2022, 142, 285-292, doi:10.1016/j.jid.2021.06.037.
- Figure 5 is not referred to in the main text; references in section 5 should probably refer to figure 5 (instead of figure 4).
Thank you for pointing out the figure that was not referenced in the manuscript. I have now added 'Figure 5' in Section 5 (Lines 178 and Line 183, page 4).
